# Flexible utilization of spatial- and motor-based codes for the storage of visuo-spatial information

**Margaret M Henderson[1,2,3]\*, Rosanne L Rademaker[4,5], John T Serences[1,4,6]**

[1]Neurosciences Graduate Program, University of California, San Diego, San Diego, United States; [2]Department of Machine Learning, Carnegie Mellon University, Pittsburgh, United States; [3]Neuroscience Institute, Carnegie Mellon University, Pittsburgh, United States; [4]Department of Psychology, University of California, San Diego, San Diego, United States; [5]Ernst Strüngmann Institute (ESI) for Neuroscience in Cooperation with Max Planck Society, Frankfurt, Germany; [6]Kavli Foundation for the Brain and Mind, University of California, San Diego, San Diego, United States

**Abstract** Working memory provides flexible storage of information in service of upcoming behavioral goals. Some models propose specific fixed loci and mechanisms for the storage of visual information in working memory, such as sustained spiking in parietal and prefrontal cortex during working memory maintenance. An alternative view is that information can be remembered in a flexible format that best suits current behavioral goals. For example, remembered visual information might be stored in sensory areas for easier comparison to future sensory inputs, or might be re-coded into a more abstract action-oriented format and stored in motor areas. Here, we tested this hypothesis using a visuo-spatial working memory task where the required behavioral response was either known or unknown during the memory delay period. Using functional magnetic resonance imaging (fMRI) and multivariate decoding, we found that there was less information about remembered spatial position in early visual and parietal regions when the required response was known versus unknown. Furthermore, a representation of the planned motor action emerged in primary somatosensory, primary motor, and premotor cortex during the same task condition where spatial information was reduced in early visual cortex. These results suggest that the neural networks supporting working memory can be strategically reconfigured depending on specific behavioral requirements during a canonical visual working memory paradigm.

**\*For correspondence:**
mmhender@cmu.edu

## Editor's evaluation

This rigorous, carefully designed and executed functional magnetic-resonance imaging study provides compelling evidence against a rigid, fixed model for how working-memory representations are maintained in the human brain. By analyzing patterns and strength of brain activity, the authors show that networks for maintaining contents in mind vary depending on the task demands and foreknowledge of anticipated responses. This manuscript will be of interest to scientists studying working memory, both in humans and in non-human primates.

## Introduction

Working memory (WM) is thought to provide a flexible mental workspace which allows organisms to hold information in mind about past experiences and use it to guide future behavior (*Atkinson and Shiffrin, 1968*; *Baddeley and Hitch, 1974*; *Jonides et al., 2016*). This system supports a wide range

of cognitive tasks, each with its own specific demands and processing constraints. Due to these varied demands, it is likely that the neural mechanisms of WM are not universal across all tasks, but adaptively adjust to the requirements of the current situation. Among the many possible differences in task requirements, one key factor is the degree to which a task encourages a perceptual versus an action-oriented coding format. For instance, in tasks that require memory for fine visual details, such as searching for a specific kind of bird based on a picture in a guidebook, the best strategy might be to represent information in a format resembling past visual inputs (a perceptual, or 'sensory-like' code). Other tasks, such as remembering a series of directions for driving to your friend's house, permit the use of multiple strategies. This driving task could be achieved using a perceptual coding strategy, such as maintaining a visuo-spatial representation of a street map. However, a better approach might be to re-code the visual information into another format, such as a series of motor plans for upcoming actions (an action-oriented, or 'motor-like' code). Such flexible re-coding of information from a more perceptual to a more action-oriented code could serve to reduce the dimensionality of representations when the correspondence between a remembered stimulus and a required action is known in advance. Thus, even tasks that are often thought to share a core component (e.g. memory in a visual format) might be accomplished via very different strategies and neural codes. Critically, past studies arguing that WM is supported by different neural loci often use tasks that differentially rely on perceptual versus action-oriented codes. As a result, evaluating claims about neural mechanisms is challenging without directly accounting for the potential impact of task demands in shaping how information is stored and used in WM.

The paradigms used in human neuroimaging experiments typically sit at the perceptual end of this continuum. In most fMRI studies of visual WM, participants are required to remember precise values of continuously varying features, and to report these remembered features using responses that cannot be pre-planned during the delay period (*Albers et al., 2013*; *Christophel et al., 2012*; *Ester et al., 2009*; *Harrison and Tong, 2009*; *Lorenc et al., 2018*; *Rademaker et al., 2019*; *Serences et al., 2009*; *Xing et al., 2013*). Such tasks may encourage top-down recruitment of the same early sensory areas that support high-precision perceptual representations (*Awh and Jonides, 2001*; *Gazzaley and Nobre, 2012*; *Pasternak and Greenlee, 2005*; *Serences, 2016*; *Sreenivasan et al., 2014*). Consistent with this idea of sensory recruitment, most studies find that patterns of voxel activation measured in visual cortex encode information about specific feature values held in memory, supporting the role of visual cortex in maintaining detailed visual representations during WM.

However, different neural loci are often implicated using visual WM tasks that enable action-oriented codes. One example is a memory-guided saccade task employed to study spatial WM using non-human primate (NHP) electrophysiological approaches. In these tasks, the position of a briefly presented cue is remembered, but that position is also predictive of the saccade that must be made at the end of the trial. Thus, the animal could solve the task by re-coding information from a spatial code into a motor code. Single unit recordings made during these tasks suggest an important role for the prefrontal cortex (PFC) in maintaining remembered information across brief delay periods (*Funahashi et al., 1989*; *Fuster and Alexander, 1971*; *Goldman-Rakic, 1995*). Consistent with these findings, others have suggested that action-oriented WM in general may rely more heavily on areas involved in planning and motor production, and less on early sensory cortex (*Boettcher et al., 2021*; *Curtis et al., 2004*; *Myers et al., 2017*). However, not all past studies have explicitly controlled the possibility for response-related remapping of information, making the boundary between perceptual and action-oriented codes unclear.

Based on the work discussed above, visual WM may be implemented quite flexibly, as opposed to having a singular neural locus or mechanism. Consistent with this idea, there are indications that even information related to the same stimulus might be stored at several different loci and in different formats (*Iamshchinina et al., 2021*; *Lee et al., 2013*; *Rademaker et al., 2019*; *Serences, 2016*). At the same time, few experiments have directly tested whether representations of memorized features in early visual cortex are subject to task modulation. Indeed, an alternative theory of WM storage suggests that memory-related signals in early sensory areas are epiphenomenal, in which case, task goals are not expected to affect the strength of representations in this part of the brain (*Leavitt et al., 2017*; *Xu, 2020*). Thus, our understanding of how behavioral requirements influence the memory networks and the codes that route information between early sensory and other brain regions is

incomplete. In particular, it is not yet clear the extent to which perceptual and action-oriented codes may be jointly employed within the context of a single WM paradigm.

Here we study visuo-spatial WM to determine if the neural mechanisms supporting information storage are flexible in the face of changing behavioral requirements, or if, alternatively, the same mechanisms are recruited irrespective of the ability to employ different strategies. Participants performed a visuo-spatial WM task in which the required behavioral response on each trial was either not known until the end of the memory delay ('uninformative' condition), or known in advance of the memory delay ('informative' condition; *Figure 1A*). This manipulation was intended to promote the use of perceptual ('sensory-like', spatial) and action-oriented ('motor-like') memory codes, respectively. We then compared representations of perceptual and action-related mnemonic information in early visual, parietal, and sensorimotor cortical regions.

To preview, we found that information about remembered spatial positions in retinotopic visual cortex decreased when the required response was known in advance, in accordance with a decreased reliance on visual cortex for information storage when action-oriented coding could be utilized. Furthermore, this decrease was accompanied by the emergence of an action-oriented memory representation in somatosensory, motor, and premotor cortex. Cross-generalization of decoding from independent model-training tasks further supports a shift from 'sensory-like' to 'motor-like' when comparing the two task conditions. These results demonstrate that the neural networks supporting WM – even in the context of a single paradigm often used to study visuo-spatial WM – can be strategically reconfigured depending on task requirements.

## Results

While in the magnetic resonance imaging (MRI) scanner, participants performed a spatial WM task in which they were required to remember the spatial position of a briefly presented target dot across a 16-s delay period. The small white target dot could appear anywhere along an invisible circle with radius of 7°. After the delay, participants reported the dot's position by comparing their memory to a response probe – a disk with two halves (light and dark gray). They used their left or right index finger to indicate on which of the two halves of the disk the target dot had been presented (*Figure 1A*). We manipulated participants' ability to pre-plan their motor action by presenting a 'preview' disk at the beginning of the delay period. The preview disk was preceded by a cue indicating whether the trial was part of the 'informative' or 'uninformative' condition. On informative trials, the preview disk matched the response disk, allowing participants to anticipate their button press at the end of the delay. On uninformative trials, the preview disk orientation was random with respect to that of the response disk, requiring that participants maintain a precise representation of the target dot position. Informative and uninformative trials were randomly intermixed throughout each run.

Task performance was overall better on trials where participants could plan their motor action in advance compared with trials when they could not (*Figure 1B*). Participants were significantly faster in the informative condition (informative mean ± SEM: 0.57 ± 0.03 s; uninformative: 1.08 ± 0.06 s; $t_{(5)}$ = –9.618; p<0.001) and also more accurate in the informative condition (informative mean ± SEM: 93.92 ± 2.12%; uninformative: 89.83 ± 1.12%; $t_{(5)}$=3.463; p=0.018). These behavioral benefits suggest that participants used the preview disk to pre-plan their motor action in the informative condition.

Next, we examined the average fMRI responses in both visual and sensorimotor cortical areas, which were each independently localized (see *Methods, Identifying regions of interest*). Our visual ROIs were retinotopically defined areas in occipital and parietal cortex, and our sensorimotor ROIs were action-related regions of primary somatosensory cortex (S1), primary motor cortex (M1), and premotor cortex (PMc). We used linear deconvolution to calculate the average hemodynamic response function (HRF) for voxels in each ROI during each task condition (see *Methods, Analysis: Univariate*; *Dale, 1999*; *Dale and Buckner, 1997*).

As expected, the BOLD signal in all retinotopic ROIs increased following visual stimulation (*Figure 1D*, left two panels; *Figure 1—figure supplement 1*). We also replicated the typical finding that occipital retinotopic areas V1–hV4 do not show sustained BOLD activation during the memory delay period (*Offen et al., 2009*; *Riggall and Postle, 2012*; *Serences et al., 2009*; *Harrison and Tong, 2009*) in either of our two task conditions. Also as expected, mean BOLD responses in parietal areas IPS0-3 showed elevated late delay period activation in the uninformative condition relative to the informative condition. This pattern is consistent with the use of a spatial code in the uninformative

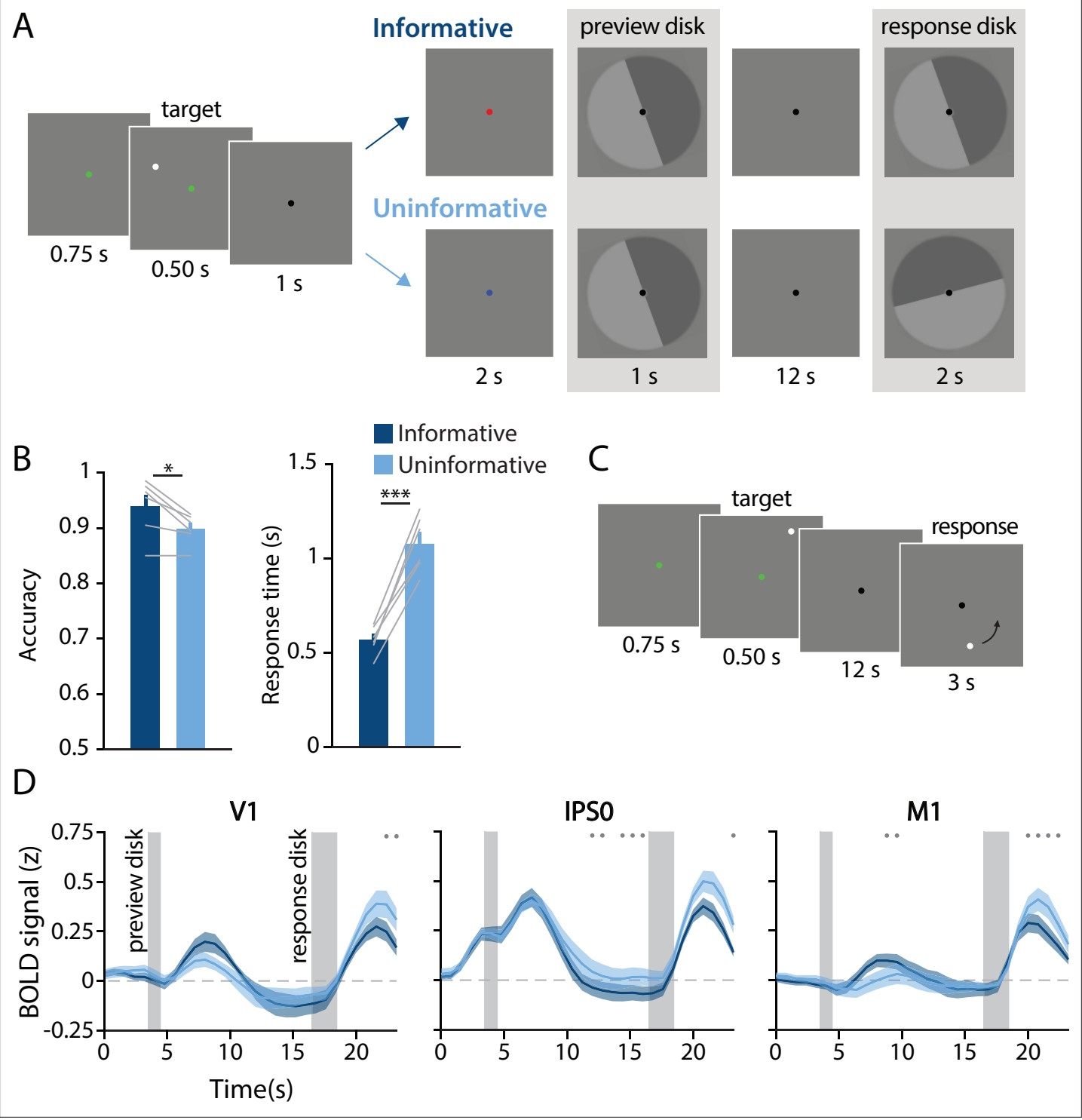

**Figure 1.** Task design, behavioral results, and univariate results. (**A**) During each trial of the main working memory task, participants remembered the spatial position of a target dot presented at a random angular position (7° eccentricity from fixation). After a 16 s delay, participants made a binary response to indicate which half of a 'response' disk the target dot position had been presented on. At the beginning of the delay, a 'preview' disk was shown that either exactly matched the response disk (top row; 'informative') or had a random orientation relative to the response disk (bottom row; 'uninformative'). In the example trial depicted, a participant in the informative condition would have pressed the button corresponding to the lighter gray side of the disk. By contrast, in the uninformative condition a participant would have pressed the button corresponding to the dark gray side, as only the final response disk was relevant to their response (see *Methods, Task: Main working memory* for more details). (**B**) Average behavioral accuracy (left) and response time (right) in the informative and uninformative conditions (individual participants are shown with gray lines). Error bars

*Figure 1 continued on next page*

*Figure 1 continued*

represent ±1 SEM across participants. Significance of condition differences was computed using paired t-tests (a single asterisk indicates p<0.05, and three asterisks indicate p<0.001). (**C**) In a separate spatial working memory mapping task, participants remembered a target dot position for 12 s and responded by moving a probe dot around an invisible circle to match the remembered position (see *Methods, Task: Spatial working memory mapping*). This mapping task was used to generate an independent dataset to train decoding models (see *Methods, Analysis: Spatial position decoding*). (**D**) Univariate hemodynamic response functions in three representative regions of interest from early visual cortex (V1), parietal cortex (IPS0), and motor cortex (M1) during the informative (dark blue) and uninformative (light blue) conditions of the main working memory task. Timepoint zero indicates the onset of the memory target. Shaded gray rectangles indicate the time periods when the 'preview' disk was onscreen (3.5–4.5 s) and when the response disk was onscreen (16.5–18.5 s). Shaded error bars represent ±1 SEM across participants. Gray dots indicate timepoints showing a significant condition difference (evaluated using a Wilcoxon signed-rank test with permutation, all p-values <0.05; see *Methods, Analysis: Univariate* for details). This plot shows three representative ROIs, see *Figure 1—figure supplement 1* for data from all ROIs.

The online version of this article includes the following figure supplement(s) for figure 1:

**Figure supplement 1.** Hemodynamic response function in each region of interest (ROI) during the informative (dark blue) and uninformative (light blue) conditions, full set of ROIs.

**Figure supplement 2.** Univariate responses in sensorimotor regions of interest (ROIs), separated by which finger was used to make a behavioral response.

condition and a motor code in the informative condition (*Curtis and D'Esposito, 2003*; *D'Esposito, 2007*; *Ester et al., 2015*; *Riggall and Postle, 2012*). Sensorimotor areas S1, M1, and PMc showed a condition difference in the opposite direction, with higher mean BOLD responses in the informative compared with the uninformative condition at several timepoints early in the delay period, consistent with an increased reliance on an action-oriented memory code (*Calderon et al., 2018*; *Donner et al., 2009*). Note that these univariate results include voxels from both hemispheres. As expected, reliable interhemispheric differences can also be observed in sensorimotor areas, with higher activation in the hemisphere contralateral to the planned button press in the informative condition (*Figure 1—figure supplement 2*).

The univariate results described above are consistent with the idea of an information 'handoff' between cortical regions as a function of perceptual and action-oriented task strategies. This brings up the important question of how such a task-related shift is reflected in population-level representations across the brain, and whether it coincides with a shift from a more sensory-like spatial memory code to a more motor-like memory code. To evaluate this question, we used a multivariate linear classifier to decode the angular spatial position of the remembered dot, based on multivoxel activation patterns measured in each ROI during the delay period (*Figure 2*). By assessing spatial decoding accuracy in each ROI, we could look at the extent to which the underlying neural code reflected a sensory-like visual representation of memorized spatial position.

To facilitate an independent comparison of decoding accuracy between the informative and the uninformative task conditions in the main WM task, we used data from an independent spatial WM task to train the classifier (*Sprague et al., 2019*; *Figure 1C*; see *Methods, Task: Spatial working memory mapping*). Before performing the decoding analysis, we subtracted from each single-trial activation pattern the mean across voxels on the same trial. This was done to ensure that any condition-specific changes in the mean BOLD responses did not contribute to differences in classification accuracy (see *Methods, Analysis: Spatial position decoding*). We then sorted the continuous angular positions into eight non-overlapping bins and used a decoding scheme with four binary classifiers, where each binary classifier was independently trained to discriminate between spatial positions that fell into bins separated by 180° (see *Figure 2A* and *Rademaker et al., 2019*). The final decoding accuracy for each task condition reflects the average decoding accuracy across all four of these binary classifiers, where chance is 50% (see *Methods, Analysis: Spatial position decoding*).

The results of this multivariate analysis demonstrate that across the uninformative and the informative conditions, spatial decoding accuracy was strongest in early visual areas V1, V2, V3, and V3AB, and became progressively weaker at more anterior regions of the visual hierarchy (*Figure 2B*). Spatial decoding accuracy was at chance in the three sensorimotor areas S1, M1, and PMc. Importantly, there was also a pronounced effect of task condition (*Figure 2B and C*). Decoding accuracy in most retinotopic areas was significantly higher for the uninformative condition, in which participants were forced to rely on a spatial memory code, compared to the informative condition, in which participants could convert to an action-oriented memory for the button press required at the end of the delay (two-way

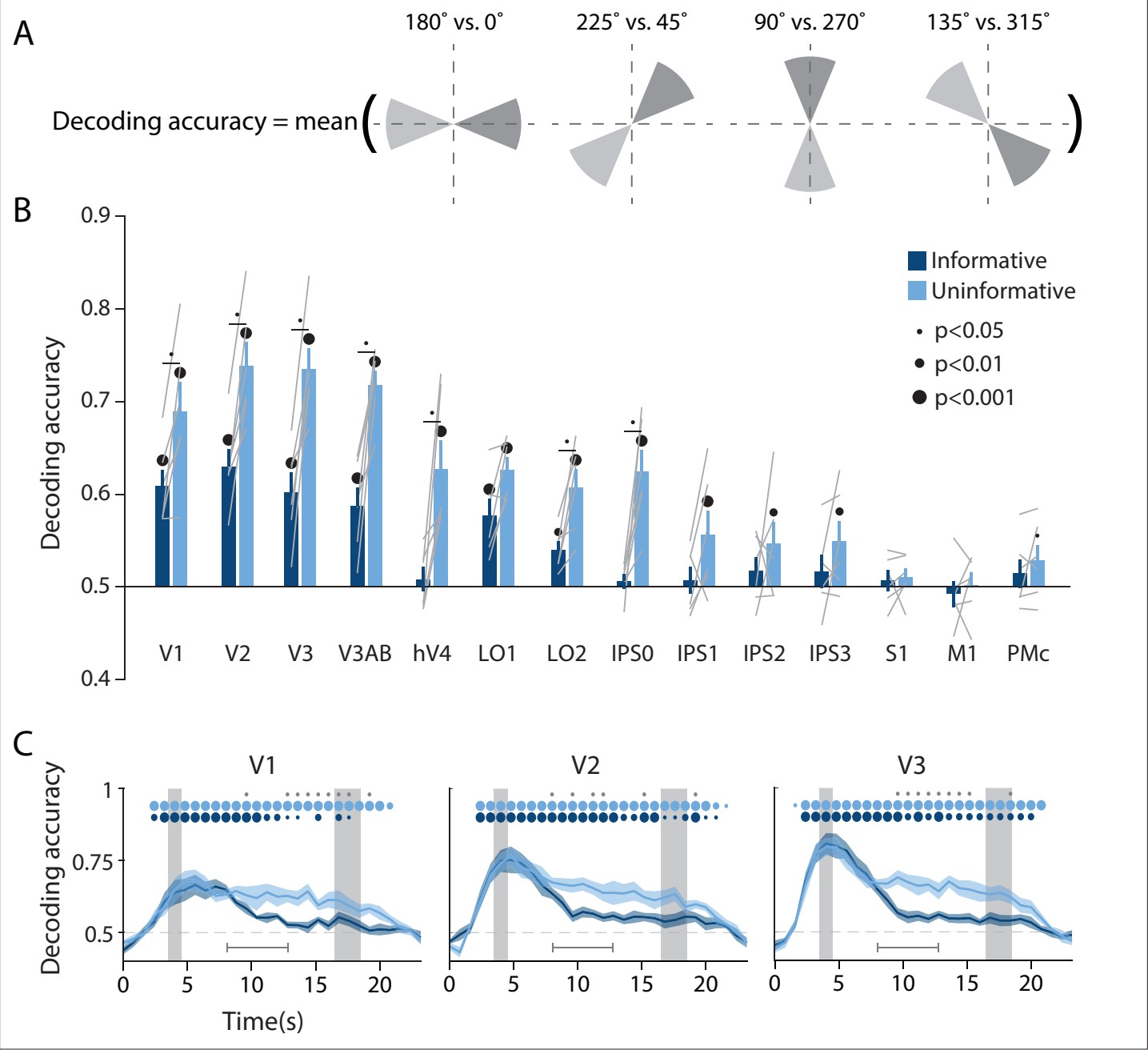

**Figure 2.** Visual and parietal areas represent spatial position more strongly during trials that require sensory-like spatial memory than during trials that allow re-coding into action-oriented memory. (**A**) Schematic of the decoding procedure. Continuous values of angular position were divided into eight discrete bins, and four binary decoders were trained to discriminate between patterns corresponding to bins 180° apart. The final decoding accuracy was the average accuracy over these four binary decoders. (**B**) Decoding accuracy for each region of interest (ROI) and task condition. The spatial decoder was always trained on data from the delay period of an independent spatial working memory mapping task (*Figure 1C*), and tested on data from the delay period of the main working memory task (averaged within a window 8–12.8 s from start of trial; see *Methods, Analysis: Spatial position decoding* for more details). Error bars reflect ±1 SEM across participants, and light gray lines indicate individual participants. Dots above bars and pairs of bars indicate the level of statistical significance within each condition, and between conditions, respectively (two-tailed p-values obtained using a Wilcoxon signed-rank test with permutation testing, see *Methods, Analysis: Spatial position decoding*). Dot sizes reflect significance level. (**C**) Spatial decoding accuracy over time in three example ROIs. Timepoint zero indicates the target onset time. Shaded gray rectangles indicate the periods of time when the 'preview' (3.5–4.5 s) and 'response' (16.5–18.5 s) disks were onscreen. Shaded error bars represent ±1 SEM across participants, colored dots indicate significance of decoding within each condition, and gray dots indicate significant condition differences, with dot sizes reflecting significance levels as in B. Gray brackets just above the x-axis in (**C**) indicate the time range over which data were averaged to produce (**B**) (i.e. 8–12.8 s). See *Figure 2—figure supplement 1* for time-resolved spatial decoding in all visual and motor ROIs.

*Figure 2 continued on next page*

*Figure 2 continued*

The online version of this article includes the following figure supplement(s) for figure 2:

**Figure supplement 1.** Time-resolved spatial decoding accuracy in every region of interest (ROI).

**Figure supplement 2.** Spatial decoding performance differs across conditions, even when training and testing a decoder within each task condition separately.

**Figure supplement 3.** Decoding accuracy for the orientation of the 'preview' disk stimulus (see *Figure 1A*).

repeated measures ANOVA with ROI and task condition as factors: main effect of ROI: $F_{(13,65)}$ = 24.548, p<0.001; main effect of task condition: $F_{(1,5)}$ = 35.537, p=0.001; ROI × condition interaction $F_{(13,65)}$ = 5.757, p<0.001; p-values obtained using permutation test; see *Methods, Analysis: Spatial position decoding*). Pairwise comparisons showed that spatial decoding accuracy was significantly higher in the uninformative compared to the informative condition in V1–hV4, LO2, and IPS0. In later IPS subregions (IPS1-3), spatial decoding accuracy was above chance in the uninformative memory condition, but at chance in the informative condition, without a significant difference between conditions. Finally, time-resolved analyses revealed that this difference between the conditions was not present during and immediately after encoding of the target dot, when the trial condition was not yet known. Instead, condition differences emerged approximately 5–6 s after the presentation of the preview disk and persisted until the final response disk appeared (*Figure 2C*, *Figure 2—figure supplement 1*). Importantly, the difference in spatial decoding accuracy between task conditions replicated when we trained and tested a decoder using data from within each condition of the main WM task as opposed to training on the independent spatial WM task. This indicates that the difference between task conditions was robust to a range of analysis choices (*Figure 2—figure supplement 2*; see *Methods, Analysis: Spatial position decoding*).

In addition to demonstrating the robustness of our findings, the two separate analysis approaches described above allow us to distinguish between two potential explanations for the observed condition difference. While one possibility is that visual cortex activation patterns contain *less overall information* about spatial position in the informative relative to the uninformative condition, another possibility is that the *format of the representations differs* between the two task conditions (e.g. *Lorenc et al., 2020*; *Vaziri-Pashkam and Xu, 2017*). Specifically, compared to the informative condition, the sensory-like spatial format used in the uninformative condition may have been more similar to the format of the independent spatial WM task (which also required a sensory-like strategy because the starting position of the response dot was unknown until the end of the trial; see *Figure 1C*). This similarity could have led to higher cross-generalization of the decoder to the uninformative condition, even if the overall spatial information content was similar between the two conditions in the main WM task. The fact that the condition difference persists when training and testing within condition rules this latter possibility out. Instead, our analyses support the interpretation that spatial WM representations in visual cortex are adaptively enhanced in the uninformative condition (when a sensory-like code is required) relative to the informative condition (where an action-oriented code can be used).

Is the quality of mnemonic spatial representations selectively modulated between task conditions, or might non-mnemonic factors – such as global arousal level – contribute to the observed differences in decoding accuracy? If non-mnemonic factors play a role, they should interact with the processing of all stimuli shown after the condition cue, including the 'preview disk' stimulus (see *Figure 1A*). A supplementary analysis revealed that the orientation of the preview disk (i.e. the boundary between its light- and dark-gray sides) could be decoded with above chance accuracy in several visual cortex ROIs in both conditions, but without a significant difference between conditions (*Figure 2—figure supplement 3*; see *Methods, Analysis: Preview disk orientation decoding* for details). This indicates that neural processing of the preview disk, a visual stimulus physically presented on the screen, was not modulated by task condition. Therefore, the effect of task condition on information content was specific to the spatial memory representation itself, and not due to a more global modulatory effect or difference in the signal-to-noise ratio of representations in early visual cortex.

In addition to spatial position decoding, we investigated the possibility of an action-oriented format of the mnemonic code by decoding upcoming actions. Since motor responses were always made with the left or the right index finger, we trained a binary linear classifier to predict which finger was associated with the required button-press on each trial based on delay period activation patterns in each

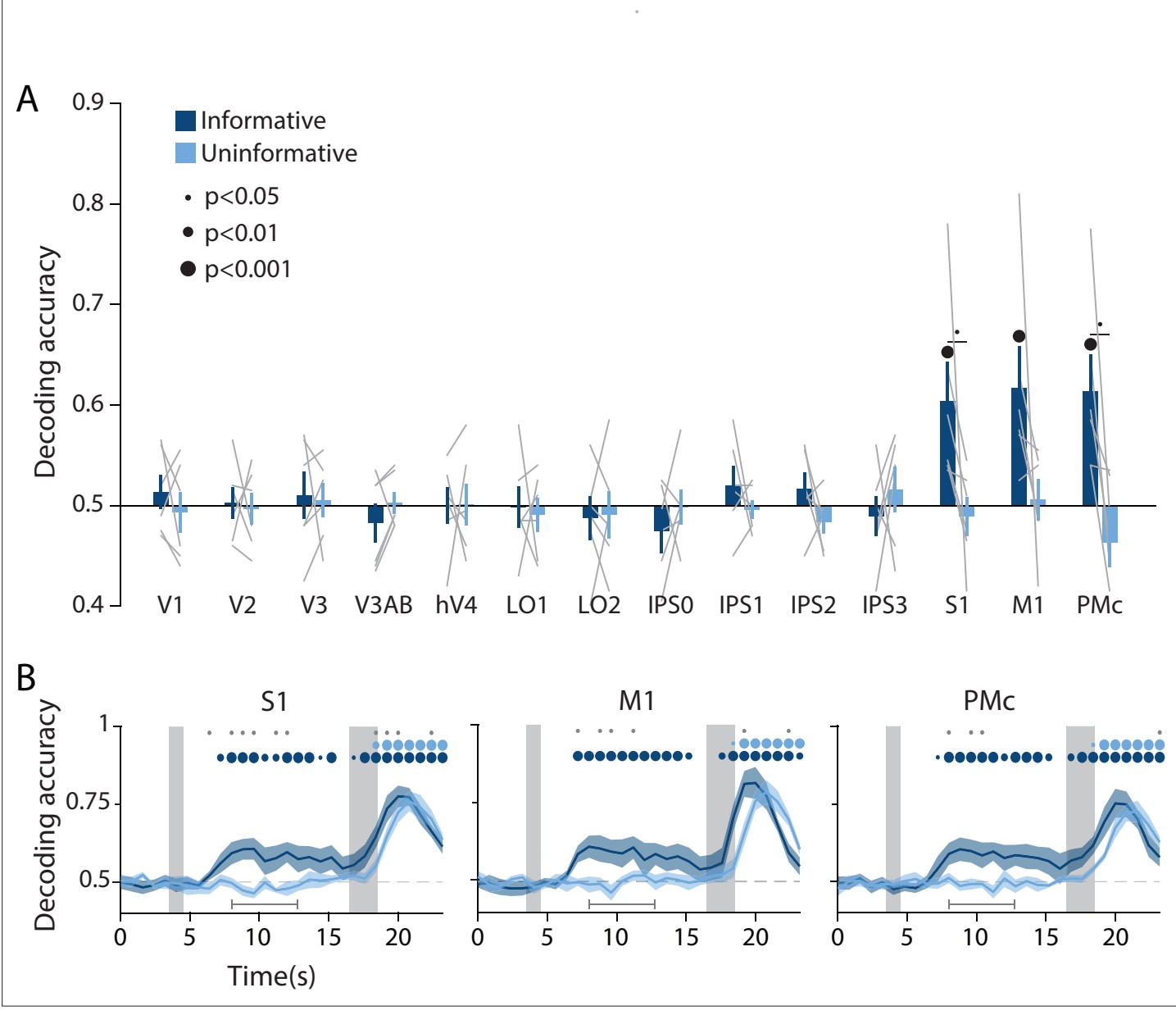

**Figure 3.** Action-oriented memory representations can be decoded from sensorimotor regions of interest (ROIs) during the delay period. (**A**) A linear decoder was trained to classify the finger (left or right index) associated with the correct motor action on each trial, using data measured during the delay period of trials in each task condition separately (averaged within a window 8–12.8 s from start of trial; see *Methods, Analysis: Action decoding* for more details). Error bars reflect ±1 SEM across participants, and light gray lines indicate individual participants. Dots above bars and pairs of bars indicate the statistical significance of decoding accuracy within each condition, and of condition differences, respectively (two-tailed p-values obtained using a Wilcoxon signed-rank test with permutation testing, see *Methods, Analysis: Action decoding*). Dot sizes reflect significance level. (**B**) Action decoding accuracy over time in three example ROIs. Timepoint zero indicates the target onset time. Shaded gray rectangles indicate the periods during which the 'preview' (3.5–4.5 s) and 'response' (16.5–18.5 s) disks were onscreen. Shaded error bars represent ±1 SEM across participants. Colored dots indicate significance of decoding accuracy within each condition, and gray dots indicate significant condition differences, with dot sizes reflecting significance levels as in A. Gray brackets just above the x-axis in (**B**) indicate the time range in which data were averaged to produce (**A**) (8–12.8 s). For time-resolved decoding in all ROIs, see *Figure 3—figure supplement 1*.

The online version of this article includes the following figure supplement(s) for figure 3:

**Figure supplement 1.** Time-resolved action decoding accuracy in every region of interest (ROI).

ROI (*Figure 3A*, see *Methods, Analysis: Action decoding* for detailed classification procedure). This analysis revealed above chance decoding of upcoming actions in each of the three sensorimotor ROIs (S1, M1, and PMc) in the informative condition but not the uninformative condition. In contrast, all retinotopic visual ROIs showed chance level action decoding for both conditions (two-way repeated measures ANOVA with ROI and task condition as factors: main effect of ROI: $F_{(13,65)} = 4.003$, $p<0.001$; main effect of condition $F_{(1,5)} = 3.802$, $p=0.106$; ROI × condition interaction: $F_{(13,65)} = 2.937$, $p=0.001$; p-values obtained using permutation test; see *Methods, Analysis: Action decoding*). A time-resolved decoding analysis revealed that, in the informative condition, information about a participants' upcoming action began to emerge approximately 4 s after the onset of the preview disk stimulus, decreased slightly toward the end of the delay period, then rose steeply after the response disk onset when the participant actually executed a motor action (*Figure 3B*, *Figure 3—figure supplement 1*). The increase in action decoding accuracy during the last part of the trial appeared sooner for the informative condition than the uninformative condition, in agreement with the speeding of behavioral response times in the informative condition (*Figure 1B*).

Together, the above results support the hypothesis that the informative and uninformative conditions differentially engaged regions of cortex and neural coding formats for information storage during WM. To characterize this difference in coding format in a more concrete way, we next leveraged two additional independent model training tasks. First, we used a spatial localizer during which participants viewed high-contrast flickering checkerboards at various spatial positions (see *Methods, Task: Spatial localizer*). With this dataset, we trained a classifier on sensory-driven visuo-spatial responses (using the same spatial position decoding method as described previously, see *Figure 2A*). We found that spatial position information generalized from this sensory-driven perceptual training data to both conditions of our main WM task (*Figure 4A*). This result supports the idea that the coding format used during spatial WM is sensory-like in nature, closely resembling the responses in visual cortex to spatially localized perceptual input. Importantly, decoding performance in early visual and parietal areas was again higher for the uninformative than the informative condition. This implies that the uninformative condition of our task resulted in a stronger and more 'sensory-like' memory code. The second independent model training task was a sensorimotor cortex localizer where participants physically pressed buttons with their left and right index fingers (see *Methods, Task: Sensorimotor cortex localizer*). We trained a decoder on this task, labeling trials according to which finger was used to physically press the buttons on each trial. We then tested this decoder on delay period activation in our main WM task. This analysis revealed above-chance decoding accuracy of the participant's upcoming action in S1, M1, and PMc, in the informative condition only (*Figure 4B*). This result supports the idea that the memory representations in the informative condition were stored in a more motor-like format which closely resembled signals measured during actual button press responses.

## Discussion

Past studies of visual WM have led to competing theories of how information is stored in the brain. Some work has emphasized the recruitment of early sensory areas, while other work has emphasized parietal and frontal association areas. Nevertheless, a general consensus exists that WM is a flexible resource (*Baddeley and Hitch, 1974*; *Gazzaley and Nobre, 2012*; *Iamshchinina et al., 2021*; *Serences, 2016*). Here we demonstrate, within a single spatial WM paradigm, that task requirements are a critical determinant of *how* and *where* WM is implemented in the brain. These data provide a partial unifying explanation for divergent prior findings that implicate different regions in visual WM (*Bettencourt and Xu, 2015*; *Ester et al., 2016*; *Iamshchinina et al., 2021*; *Rademaker et al., 2019*; *Xu, 2018*; *Xu, 2020*). More importantly, however, our data show that WM flexibly engages different cortical areas and coding formats, even in the context of a task that is commonly used to study a single underlying construct (i.e. visuo-spatial WM). These findings highlight the goal-oriented nature of WM: the brain's mechanisms for storage are not fully specified by the type of information being remembered, but instead depend on how the encoded information will be used to guide future behavior.

We used a spatial WM task wherein participants could anticipate their behavioral response ahead of the delay interval on half of the trials (*Figure 1A*), encouraging re-coding of visuo-spatial memory information to a more action-oriented representational format. On the other half of trials, participants were unable to anticipate their behavioral response, and had to rely on a sensory-like spatial memory representation. The decoding accuracy of a remembered position from activation patterns

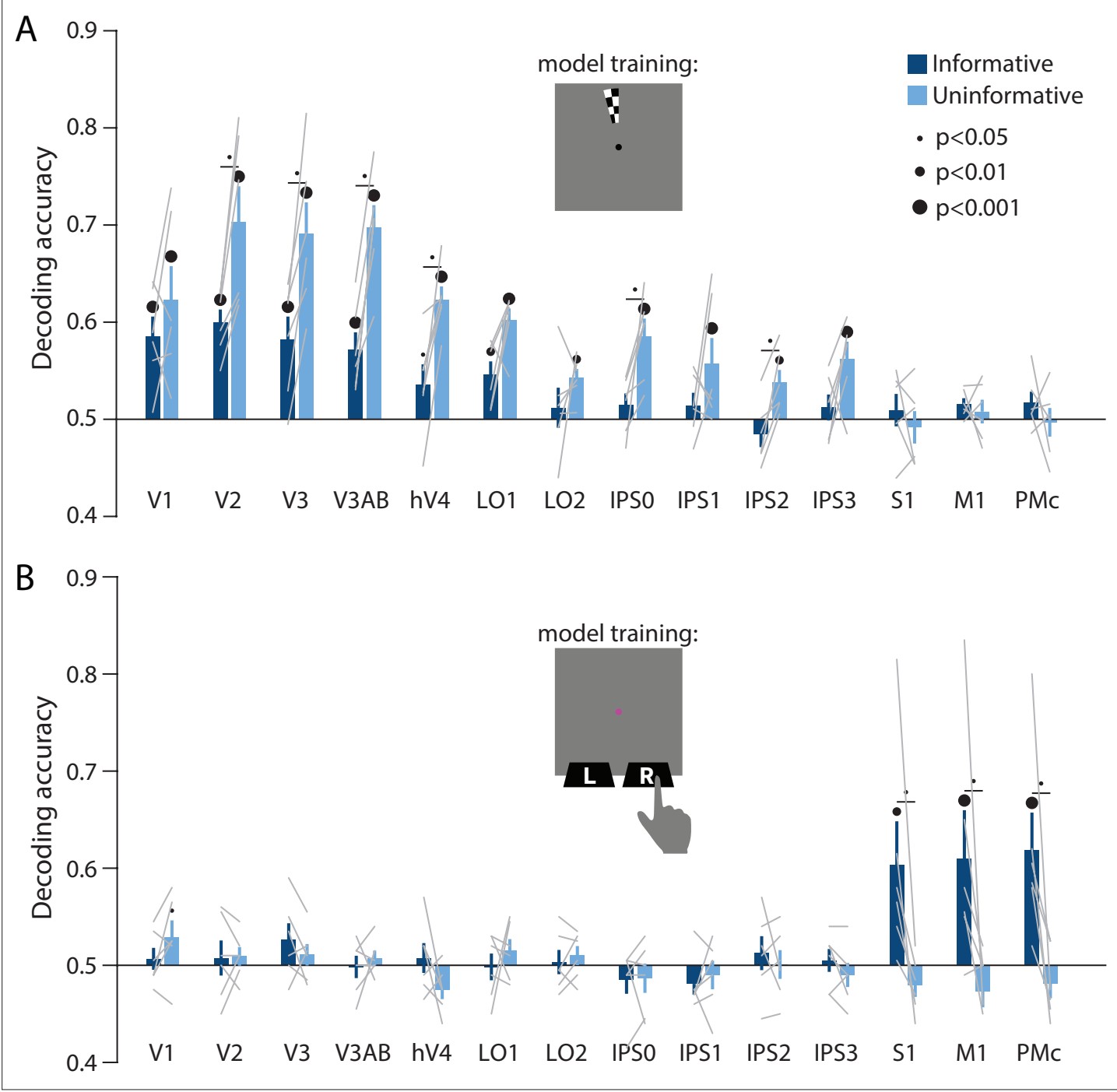

**Figure 4.** Perceptual and action-oriented memory representations generalize from signals associated with perceptual input and physical button-press responses, respectively. (**A**) Decoding accuracy of spatial memory position when training on data from an independent sensory localizer (see *Methods, Task: Spatial localizer*, and *Methods, Analysis: Spatial position decoding* for details on this task and on decoding procedure). (**B**) Decoding accuracy of action-oriented memory codes (i.e. the finger associated with the correct motor action on each trial) when training on data from an independent button pressing task (see *Methods, Task: Sensorimotor cortex localizer* and *Methods, Analysis: Action decoding* for details on this task and on decoding procedure). In both panels, error bars reflect ±1 SEM across participants, and light gray lines indicate individual participants. Dots above bars and pairs of bars indicate the statistical significance of decoding accuracy within each condition, and of condition differences, respectively, both evaluated using non-parametric statistics. Dot sizes reflect significance level. Inserts show a cartoon of the localizer tasks used to train the decoder for these analyses.

in visual cortex was lower when participants had the opportunity to anticipate their eventual motor action, compared to when they had to rely exclusively on sensory-like visual memory. This effect was consistent across a range of early visual and parietal ROIs (*Figure 2*), supporting the idea that the recruitment of visual cortex for WM storage is task dependent. Conversely, during the same task condition where visuo-spatial representations became weaker in visual cortex, sensorimotor ROIs showed above-chance decoding of the participant's planned action during the delay period (*Figure 3*). We additionally show that the format of these two memory representations was markedly different: the spatial memory code generalized from an independent task in which stimuli at various spatial positions were actively viewed, whereas the action-related memory code generalized from an independent task in which participants made actual alternating button-press responses (*Figure 4*). Together, these results demonstrate that rather than having one fixed locus or mechanism, visual WM can be supported by different loci and coding schemes that are adapted to current task requirements.

While we found widespread decoding of the contents of WM, we generally did not find sustained increases in univariate BOLD responses during the delay (*Figure 1D*; *Figure 1—figure supplement 1*). Specifically, visual areas V1–hV4 and LO2 showed no sustained BOLD responses or differences in the mean BOLD signal between conditions during the delay period, despite these areas showing condition differences in spatial decoding accuracy. This adds to an existing body of research showing a dissociation between univariate and multivariate indices of WM storage (*Emrich et al., 2013*; *Ester et al., 2015*; *Harrison and Tong, 2009*; *Riggall and Postle, 2012*; *Serences et al., 2009*). It additionally suggests that the observed differences in decoding performance cannot be explained by global changes in signal due to arousal or task difficulty, but are instead attributable to differences in the amount of information represented within population-level patterns of activation in visual cortex. Furthermore, we found that information about the preview disk (see *Figure 1A*), a physical stimulus that appeared briefly on the screen early in the delay period, was not significantly different between task conditions (*Figure 2—figure supplement 3*). This finding further highlights that the observed modulation of spatial decoding accuracy in visual cortex was specific to the memory representation, and not due to a non-specific change in signal-to-noise ratio within early visual areas.

At the same time, in multiple subregions of IPS, we did observe higher univariate delay period activation for the uninformative condition relative to the informative condition. In contrast, the opposite was true in sensorimotor areas (*Figure 1D*; *Figure 1—figure supplement 1*). This finding parallels previous work showing higher univariate delay period activation in IPS during a task where oculomotor responses were decoupled from spatial memory items (i.e. encouraging a more perceptual strategy), whereas oculomotor areas showed higher activation when responses and memoranda were yoked (i.e. encouraging a more action-oriented strategy; *Curtis et al., 2004*). This past work suggested specialized networks for perceptual 'sensory-like' versus action-oriented 'motor-like' memories. Our current demonstration of a shift from visual to sensorimotor codes further builds on this by showing how the neural systems underlying WM may be flexibly reconfigured, and memory representations flexibly reformatted, based on behavioral requirements.

While information about the remembered spatial position was substantially lower in the informative condition than the uninformative condition, decoding accuracy did not fall entirely to chance in early visual cortex during informative cue trials (*Figure 2B*). With the possible exception of area V1, this was true even when looking at decoding at the very end of the WM delay period (*Figure 2C*, *Figure 2—figure supplement 1*). Several factors may have contributed to this above chance decoding accuracy. First, it is possible that on some fraction of informative trials, participants failed to properly encode or process the condition cue, and as a result maintained a high-precision spatial representation throughout the delay period. Such lapses of attention are possible, given that the conditions in our task were randomly interleaved on a trial-by-trial basis. Second, it is possible that participants faithfully followed the informative cue, and made use of an action-oriented strategy, but some amount of visuo-spatial memory information was nonetheless maintained in visual cortex. This would suggest that the recruitment of sensory-like codes in visual cortex is at least partially obligatory when performing a visuo-spatial WM task. This account is consistent with past findings showing that items which are not immediately relevant for a task, but will be relevant later (i.e. 'unattended memory items'), can still be decoded from early visual cortex (*Iamshchinina et al., 2021*; re-analysis of data from *Christophel et al., 2018*; but see also *Lewis-Peacock et al., 2012*; *LaRocque et al., 2017*). However, this account is *inconsistent* with past findings showing that when a once-maintained item is no longer needed for

behavior, it is no longer decodable from early visual areas (*Harrison and Tong, 2009*; *Lewis-Peacock et al., 2012*; *Sprague et al., 2014*; *Sprague et al., 2016*; *Lorenc et al., 2020*). In our paradigm, we cannot rule out that – even on trials where participants could completely discard spatial memory information in favor of an action-oriented code – some amount of visuo-spatial information was still maintained in visual cortex, reflecting a partially distributed code. Some support for the idea that sensory- and action-like codes can be maintained and accessed simultaneously during a memory task comes from a study by *van Ede et al., 2019a*, who showed such simultaneity using EEG during a task where both action-related and sensory-like codes were relevant for performance. Further experiments will be needed to determine how such distributed codes may be engaged during memory tasks in general, and how the extent to which a neural code is more distributed versus more punctate may change as a function of task requirements.

We observed above-chance delay period decoding of participants' planned actions in S1, M1, and PMc in the informative condition. Each of these areas has previously been shown to exhibit prepara-tory motor activity in the context of tasks involving delayed reaching or finger pressing (*Ariani et al., 2022*; *Calderon et al., 2018*; *Cisek and Kalaska, 2005*; *Donner et al., 2009*). In our data, information regarding the upcoming action became detectable around 4 s after the onset of the preview disk – roughly the same time that spatial decoding accuracy in visual cortex dropped in the informative compared to the uninformative condition (*Figure 2C* and *Figure 3B*). While suggestive, the temporal resolution of fMRI does not allow us to draw firm conclusions about the relative timing of each process. Nevertheless, these results are broadly consistent with a theory in which action selection unfolds in parallel to task-related processing of visual input (*Cisek and Kalaska, 2010*; *Donner et al., 2009*; *Klein-Flügge and Bestmann, 2012*; *van Ede et al., 2019a*). Additionally, we found that information about upcoming actions declined toward the end of the delay period, dropping to chance just before the response disk appeared (*Figure 3B*). This time course is consistent with a transient signal related to the initial formation of an action-oriented memory representation, and aligns with recent EEG find-ings in human motor cortex following a retro-cue in a visual WM task (*Boettcher et al., 2021*).

Past work has investigated the influence of action-oriented coding on WM representations. For instance, it has been shown that high priority items, meaning those that are relevant for upcoming actions, tend to be represented more robustly than items that are not immediately relevant to behavior (*Christophel et al., 2012*; *Lewis-Peacock et al., 2012*; *Lorenc et al., 2020*; *Rose et al., 2016*; *Sprague et al., 2016*; but see also *Barbosa et al., 2021*; *Iamshchinina et al., 2021*). Prioritized WM representations can be further reconfigured so that they are optimized for future behavior, which may include the activation of circuits related to motor output (*Myers et al., 2017*; *Nobre and Stokes, 2019*; *Schneider et al., 2017*; *Souza and Oberauer, 2016*; *van Ede et al., 2019a*). These past studies of action preparation have typically used relatively coarse measures of visual information content such as decoding which of two items was currently selected. In contrast, here we measured the information about a continuous remembered feature value that was reflected in patterns of activation in visual cortex. We simultaneously showed a decrease in spatial information in visual cortex and an increase in action-related information in sensorimotor cortex when an action-oriented mnemonic format was encouraged. This finding solidifies the evidence showing that information may shift between qualita-tively different kinds of codes as a function of behavioral requirements, as opposed to just waxing and waning over time within the same format/brain areas.

Past work on saccadic eye movements further supports the link between the neural circuits that support WM and those that support action preparation and execution. For instance, prior work has demonstrated that when retrieving or selecting an item within WM, eye movements exhibit a system-atic bias toward the spatial position at which the item was originally presented, even without a phys-ical stimulus to guide eye movements (*Spivey and Geng, 2001*; *Ferreira et al., 2008*; *van Ede et al., 2019b*; *van Ede et al., 2020*). These eye movements may play a functional role in retrieving items from WM (*Ferreira et al., 2008*; *van Ede et al., 2019b*) and can index processes such as attentional selec-tion within WM (*van Ede et al., 2019b*; *van Ede et al., 2020*). In other work, the interaction between memory and action has been shown in the opposite direction, where the prioritization of a feature or location for motor actions can lead to prioritization in WM (*Heuer et al., 2020*). For example, when participants make a saccade during the delay period of a visual WM task, memory performance is enhanced for items originally presented at the saccade target location (*Ohl and Rolfs, 2020*). This effect occurs even when saccade target locations conflict with top-down attentional goals, suggesting

it reflects an automatic mechanism for selection based on action planning (*Ohl and Rolfs, 2020*; *Heuer et al., 2020*). Finally, the planning of both memory-guided saccades and anti-saccades is associated with topographically specific activation in early visual cortex measured with fMRI, suggesting that neural representations of upcoming actions are coded similarly to representations used for WM storage (*Saber et al., 2015*; *Rahmati et al., 2018*). Together, these findings suggest an important functional association between WM and motor planning within the domain of eye movement control. Our work expands upon this idea by demonstrating the engagement of motor circuits for the maintenance of action-oriented WM representations using a paradigm outside the realm of eye movement planning. Additionally, while prior work suggested a close correspondence between perceptual and action-oriented WM codes, our work demonstrates that these two types of representations can also be strategically dissociated depending on task demands and the nature of the motor response being planned (i.e. a manual response versus a saccadic response).

Finally, beyond the impact of more perceptual and action-oriented WM strategies, other aspects of task requirements have been shown to influence the neural correlates of WM. For example, one experiment found that information about visual objects could be decoded from extrastriate visual areas on blocks of trials that required memory for visual details, but from PFC on blocks of trials where only the object's category had to be maintained (*Lee et al., 2013*). Similarly, another experiment showed that instructing participants to maintain items in a visual, verbal, or semantic code resulted in format-specific patterns of neural activation measured with fMRI (*Lewis-Peacock et al., 2015*). These findings, along with our results, further support a framework in which behavioral requirements strongly influence the format of WM representations.

In our action-decoding analyses (*Figure 3* and *Figure 4B*), we interpret the decoded representations as primarily reflecting the planning of upcoming actions and maintenance of an action plan in WM, rather than reflecting overt motor actions. This interpretation is justified because participants were explicitly instructed to withhold any physical finger movements or button presses until the end of the delay period, when the probe disk appeared. We observed participants throughout their behavioral training and scanning sessions to ensure that no noticeable finger movements were performed. Nevertheless, we cannot rule out the possibility that some amount of low-grade motor activity (such as a subtle increase in pressure on the button box, or increased muscle tone in the finger) occurred during the delay period in the informative condition and contributed to our decoding results. Further experiments incorporating a method such as electromyography recordings from the finger muscles would be necessary to resolve this. At the same time, this possibility of low-grade motor activity does not substantially change the interpretation of our results, namely that the representation of upcoming actions in motor, premotor, and supplementary motor cortex reflects the maintenance of a prospective action plan within WM. As detailed in the preceding paragraphs, a large body of work has drawn a link between action preparation and WM in the brain, with motor circuits often actively engaged by the selection and manipulation of items within WM, and motor actions in turn influencing the quality of memory representations. Based on this framework, the action-oriented representations in our task, independent of whether they may reflect a contribution from physical motor activity, are within the range of representations that can be defined as WM.

In contrast to human neuroimaging experiments, which often find evidence for sensory recruitment (i.e. WM decoding from early sensory cortices), research with NHPs has generally found less evidence for maintenance of feature-specific codes in early sensory regions. The difference in results between human and primate studies may be partially accounted for by differences in measurement modality. For example, the BOLD response is driven in part by modulations of local field potentials (*Boynton, 2011*; *Goense and Logothetis, 2008*; *Logothetis et al., 2001*; *Logothetis and Wandell, 2004*), which may contain information about remembered features, such as motion direction, even when single unit spike rates do not (*Mendoza-Halliday et al., 2014*). Part of the discrepancy may also be related to differences in tasks used in human studies versus NHP studies. For example, some NHP studies use tasks that allow for motor coding strategies, such as the traditional memory-guided saccade paradigm (*Funahashi et al., 1989*; *Fuster and Alexander, 1971*; *Goldman-Rakic, 1995*). At the same time, other NHP experiments have dissociated sensory and action-related information (*Funahashi et al., 1993*; *Mendoza-Halliday et al., 2014*; *Miller et al., 1996*; *Panichello et al., 2019*), suggesting that the difference between perceptual and action-oriented coding alone cannot account for differences between studies. Another aspect of task design that may contribute to these differences is that NHP

tasks often require animals to remember one of a small, discrete set of stimuli (e.g. *Funahashi et al., 1993*; *Mendoza-Halliday et al., 2014*; *Miller et al., 1996*). This may encourage a more discretized categorical representation rather than a highly detailed representation, which may rely less strongly on the finely tuned neural populations of early sensory areas (*Lee et al., 2013*). For example, some NHP electrophysiology studies have shown spiking activity in V1 that reflects the contents of WM using tasks that require precise spatial representations that may not be easily re-coded into a non-sensory format (e.g. the curve-tracing task used by Van Kerkoerle and colleagues; *Supèr et al., 2001*; *van Kerkoerle et al., 2017*).

Together, our findings suggest that the neural mechanisms that underlie WM are dynamic and can be flexibly adjusted, even in the context of a single paradigm typically used to study visuo-spatial WM. When our participants were given the possibility to switch from a purely visual to a motor-based WM code, the amount of information regarding precise spatial position dropped in early visual and parietal cortex, while information about an upcoming action became decodable from sensorimotor cortex. This experiment highlights how in a single paradigm, we can measure multiple dissociable mechanisms supporting WM. More broadly, these results open the door for future experiments to explore other task factors that may dynamically alter how WM is represented in the brain, and further push the boundaries of our knowledge about WM and cognitive flexibility.

## Materials and methods

### Participants

Six participants (two male) between the ages of 20 and 34 were recruited from the University of California, San Diego (UCSD) community (mean age 27.2 ± 2.7 years). All had normal or corrected-to-normal vision. One additional participant (male) participated in an early pilot version of the study, but is not included here as they did not complete the final experiment. The study protocol was approved by the Institutional Review Board at UCSD, and all participants provided written informed consent. Each participant performed a behavioral training session lasting approximately 30 min, followed by three or four scanning sessions, each lasting approximately 2 hr. Participants were compensated at a rate of $10 /hr for behavioral training and $20 /hr for the scanning sessions. Participants were also given 'bonus' money for correct performance on certain trials in the main behavioral task (see *Task: Main working memory*), up to a maximum of $40 bonus.

### Magnetic resonance imaging (MRI)

All MRI scanning was performed on a General Electric (GE) Discovery MR750 3.0T research-dedicated scanner at the UC San Diego Keck Center for Functional Magnetic Resonance Imaging. Functional echo-planar imaging (EPI) data were acquired using a Nova Medical 32-channel head coil (NMSC075-32-3GE-MR750) and the Stanford Simultaneous Multislice (SMS) EPI sequence (MUX EPI), with a multi-band factor of 8 and 9 axial slices per band (total slices = 72; 2 mm$^3$ isotropic; 0 mm gap; matrix = 104 × 104; field of view (FOV) = 20.8 cm; repetition time/time to echo (TR/TE) = 800/35 ms; flip angle = 52°; inplane acceleration = 1). Image reconstruction and un-aliasing procedures were performed on servers hosted by Amazon Web Services, using reconstruction code from the Stanford Center for Neural Imaging. The initial 16 TRs collected at sequence onset served as reference images required for the transformation from k-space to image space. Two short (17 s) 'topup' datasets were collected during each session, using forward and reverse phase-encoding directions. These images were used to estimate susceptibility-induced off-resonance fields (*Andersson et al., 2003*) and to correct signal distortion in EPI sequences using FSL topup functionality (*Jenkinson et al., 2012*).

In addition to the experimental scanning sessions, each participant participated in a separate retinotopic mapping session during which we also acquired a high-resolution anatomical scan. This anatomical T1 image was used for segmentation, flattening, and delineation of the retinotopic mapping data. For four out of the six participants, the anatomical scan was obtained using an in vivo eight-channel head coil with accelerated parallel imaging (GE ASSET on a FSPGR T1-weighted sequence; 1 × 1 × 1 mm$^3$ voxel size; 8136 ms TR; 3172 ms TE; 8° flip angle; 172 slices; 1 mm slice gap; 256 × 192 cm matrix size), and for the remaining two participants this scan was collected using the same 32-channel head coil used for functional scanning (anatomical scan parameters used with 32-channel coil were identical to those used with the eight-channel coil). Anatomical scans collected with the 32-channel

head coil were corrected for inhomogeneities in signal intensity using GE's 'phased array uniformity enhancement' method.

## Pre-processing of MRI data

All pre-processing of MRI data was performed using software tools developed and distributed by FreeSurfer and FSL (available at https://surfer.nmr.mgh.harvard.edu and http://www.fmrib.ox.ac.uk/fsl). First, we used the recon-all utility in the FreeSurfer analysis suite (*Dale, 1999*) to perform cortical surface gray-white matter volumetric segmentation of anatomical T1 scans. The segmented T1 data were used to define cortical meshes on which we specified retinotopic ROIs used for subsequent analyses (see *Identifying ROIs*). T1 data were also used to align multisession functional data into a common space: for each of the experimental scan sessions, the first volume of the first run was used as a template to align the functional data from that session to the anatomical data. Co-registration was performed using FreeSurfer's manual and automated boundary-based registration tools (*Greve and Fischl, 2009*). The resulting transformation matrices were then used to transform every four-dimensional functional volume into a common space, using FSL FLIRT (*Jenkinson et al., 2002*; *Jenkinson and Smith, 2001*). Next, motion correction was performed using FSL MCFLIRT (*Jenkinson et al., 2002*), without spatial smoothing, with a final sinc interpolation stage, and 12° of freedom. Finally, slow drifts in the data were removed using a high-pass filter (1/40 Hz cutoff). No additional spatial smoothing was performed.

The above steps were performed for all functional runs, including the main WM task (see *Task: Main working memory*), spatial WM mapping (see *Task: Spatial working memory mapping*), sensorimotor cortex localizer (see *Task: Sensorimotor cortex localizer*), and spatial localizer (see *Task: Spatial localizer*). Following this initial pre-processing, for all run types, we normalized the time series data by z-scoring each voxel's signal across each entire scan run (this and all subsequent analyses were performed in Matlab 2018b; see *Henderson, 2022* for our code). Deconvolution (see *Analysis: Univariate*) of main task data was performed on this continuous z-scored data. Next, we epoched the data based on the start time of each trial. Since trial events were jittered slightly with respect to TR onsets, we rounded trial start times to the nearest TR. This epoched data was used for time-resolved decoding analyses (*Figure 2C* and *Figure 3B*, *Figure 2—figure supplement 1*, and *Figure 3—figure supplement 1*). For the time-averaged analyses (*Figure 2B*, *Figure 3A*, *Figure 4*, *Figure 2—figure supplement 2*), we obtained a single estimate of each voxel's response during each trial. For the main task, we obtained this value using an average of the timepoints from 10 to 16 TRs (8–12.8 s) after trial onset, which falls in the delay period of the task. For the spatial WM mapping task, we used an average of the timepoints from 6 to 12 TRs (4.8–9.6 s) after trial onset, which falls in the delay period of this task. For the spatial localizer task, we averaged over the timepoints from 4 to 7 TRs (3.2–5.6 s) after stimulus onset. For the button-pressing task (see *Task: Sensorimotor cortex localizer*) we averaged over the timepoints from 4 to 7 TRs (3.2–5.6 s) after trial onset. The data from these latter two tasks was additionally used to identify voxels based on their spatial selectivity and action selectivity, respectively (see *Identifying ROIs*).

## Identifying regions of interest (ROIs)

We followed previously published retinotopic mapping protocols to define the visual areas V1, V2, V3, V3AB, hV4, IPS0, IPS1, IPS2, and IPS3 (*Engel et al., 1997*; *Jerde and Curtis, 2013*; *Sereno et al., 1995*; *Swisher et al., 2007*; *Wandell et al., 2007*; *Winawer and Witthoft, 2015*). Participants performed mapping runs in which they viewed a contrast-reversing black and white checkerboard stimulus (4 Hz) that was configured as either a rotating wedge (10 cycles, 36 s/cycle), an expanding ring (10 cycles, 32 s/cycle), or a bowtie (8 cycles, 40 s/cycle). To increase the quality of retinotopic data from parietal regions, participants performed a covert attention task on the rotating wedge stimulus, which required them to detect contrast dimming events that occurred occasionally (on average, one event every 7.5 s) in a row of the checkerboard (mean accuracy = 74.4 ± 3.6%). The maximum eccentricity of the stimulus was 9.3°.

After mapping the individual retinotopic ROIs for each participant, we used data from our spatial localizer task (see *Task: Spatial localizer*) to identify voxels within each retinotopic ROI that were selective for the region of visual space in which our spatial memory positions could appear. Data from this localizer task were analyzed using a general linear model (GLM) implemented in FSL's FEAT (FMRI

Expert Analysis Tool, version 6.00). Brain extraction (*Smith, 2002*) and pre-whitening (*Woolrich et al., 2001*) were performed on individual runs before analysis. Predicted BOLD responses for each of a series of checkerboard wedges were generated by convolving the stimulus sequence with a canonical gamma hemodynamic response (phase = 0 s, s.d. = 3 s, lag = 6 s). Individual runs were combined using a standard weighted fixed effects analysis. For each of the 24 possible wedge positions, we identified voxels that were significantly more activated by that position than by all other positions (p<0.05, false discovery rate corrected). We then merged the sets of voxels that were identified by each of these 24 tests and used this merged map to select voxels from each retinotopic ROI for further analysis.

In addition to mapping visual ROIs, we also mapped several sensorimotor ROIs. We did this by intersecting data from a simple button-press task (see *Task: Sensorimotor cortex localizer*) with anatomical definitions of motor cortex. Data from the sensorimotor localizer were analyzed using a GLM in FSL's FEAT, as described above for the spatial localizer. Predicted BOLD responses for left- and right-handed button presses were generated by convolving the stimulus sequence with a gamma HRF (phase = 0 s, s.d. = 3 s, lag = 5 s). We identified voxels that showed significantly more activation for the contralateral index finger than the ipsilateral index finger (p<0.05, false discovery rate corrected). This procedure was done separately within each hemisphere. We then defined each sensorimotor ROI by intersecting the map of above-threshold voxels with the anatomical definitions of Brodmann's areas (BAs) identified by FreeSurfer's recon-all segmentation procedure (*Dale et al., 1999*; *Fischl et al., 2008*). Specifically, the functionally defined mask was intersected with BA 6 to define premotor cortex (PMc), with BA 4 to define primary motor cortex (M1), and with BA 1, 2, and 3 combined to define primary somatosensory cortex (S1) (*Brodmann, 1909*; *Fulton, 1935*; *Penfield and Boldrey, 1937*). Final sizes of all visual and sensorimotor motor ROIs are reported in *Supplementary file 1*.

## Task: Main working memory

For all tasks described here, stimuli were projected onto a screen 21.3 cm wide × 16 cm high, fixed to the inside of the scanner bore just above the participant's chest. The screen was viewed through a tilted mirror attached to the headcoil, from a viewing distance of 49 cm. This resulted in a maximum vertical extent (i.e. bottom to top) of 18.5°, and a maximum vertical eccentricity (i.e. central fixation to top) of 9.3°. The background was always a mid-gray color, and the fixation point was always a black circle with radius 0.2°. All stimuli were generated using Ubuntu 14.04, Matlab 2017b, and the Psychophysics toolbox (*Brainard, 1997*; *Kleiner et al., 2007*).

During runs of the main WM task, the overall task that participants performed was to remember the position of a small target dot, maintain it across a delay period, and then report which side of a spatial boundary the remembered position had fallen on (*Figure 1A*). Each trial began with the fixation point turning green for 750 ms, to alert the participant that the spatial memory target was about to appear. Next, a white target dot (radius = 0.15°) appeared for 500 ms at a pseudo-random position on an imaginary ring 7° away from fixation (details on target positions given two paragraphs down). Participants were required to remember the precise position of this target dot. After presentation of this spatial memory target, the fixation point turned back to black for 1 s, then turned either red or blue for 2 s. This color cue indicated to the participant whether the current trial was an 'informative' or 'uninformative' trial (see next paragraph for explanation of the conditions). Next, a disk stimulus appeared for 1 s. This stimulus consisted of a circle 9.7° in radius, divided into two equal halves, with each side a different shade of gray (visibly lighter and darker relative to the mean-gray background; see *Figure 1A*). The disk could be rotated about its center by an amount between 1 and 360°. To avoid the disk overlapping with the fixation point, an aperture of radius 0.4° was cut out of the middle of the disk, creating a donut-like shape. The inner and outer edges of the donut were smoothed with a 2D Gaussian kernel (size = 0.5°, sigma = 0.5°), but the boundary between the two halves of the disk was sharp. This 'preview disk' stimulus was followed by a 12 s delay period. Following the delay period, a second disk stimulus appeared for 2 s, serving as the response probe (i.e. the 'response disk'). At this point, participants responded with a button press to indicate which side of the response disk the memory target had been presented on. Specifically, they used either their left or right index finger to indicate whether the target dot fell within the light gray or dark gray side of the response disk. On each scan run, the light gray shade and dark gray shade were each associated with one response finger, and the mapping between shade of gray (light or dark) and finger (left or right index finger) was counterbalanced across sessions within each participant.

Participants were reminded of the mapping between shades of gray and response fingers at the start of each scan run.

For trials in the 'informative' condition, the orientation of the response disk was identical to that of the preview disk. For trials in the 'uninformative' condition, the orientation of the response disk was random (and unrelated to the orientation of the preview disk). Thus, for the informative condition, participants had complete knowledge of the required action as soon as the preview disk appeared, but for the uninformative condition, they had no ability to anticipate the required action. Participants were instructed not to make any physical finger movements until the response disk appeared. Trials of the two task conditions (i.e. informative/uninformative) were randomly interleaved within every scan run. At the start of each scan run, the participant was shown an instruction screen which reminded them of the color/condition mapping and the shade of gray/finger mapping that was in effect for that session. The mapping between color (red/blue) and condition was counterbalanced across participants, but fixed within a given participant.

Each run of the main task consisted of 20 trials, with each trial followed by an inter-trial interval jittered randomly within a range of 1–5 s. The total length of each run was 466 s. Participants performed 10 runs of this task per scan session, and completed a total of 20 runs (or 400 trials) across two separate scan sessions. All counterbalancing was performed at the session level: within each session, there were 100 trials of each condition (informative or uninformative), and on each of these 100 trials the memory target was presented at a unique polar angle position. Specifically, target positions were sampled uniformly along a 360° circle, with a minimum spacing of 3.6° between possible target positions. The orientation of the response disk (which determined the spatial boundary to which the memory target position was compared) also took 100 unique and uniformly spaced values along a 360° circle within each condition. The possible disk orientations were shifted by 1.8° relative to the possible spatial memory positions, so that the memory position was never exactly on the boundary of the disk. To ensure that the joint distribution of memory positions and boundary positions was close to uniform, we broke the 100 possible positions along the 360° circle into 10 bins of 10 positions each. Across all 100 trials of each condition, each combination of the bin for spatial memory position and the bin for boundary orientation was used once. For the informative condition, the preview disk always took on the same orientation as the response disk. For the uninformative condition, the preview disk was assigned a random orientation, using the same 100 orientations used for the response disk but in a random order. Finally, trials for both task conditions were randomly shuffled and split evenly into 10 runs. As a result, task condition, memory target position, and response disk orientation were balanced across each session, but not balanced within individual runs.

To encourage participants to encode the spatial positions with high precision, we rewarded participants monetarily for correct performance on 'hard' trials on which the spatial memory target was close to the boundary. These 'hard' trials were identified as those where the spatial memory item and the boundary belonged in the same bin, according to the angular position bins described above. Participants received $1 for correct performance on each 'hard' trial, for a maximum of $40. Across participants, the average bonus received was $32.83 ± 2.86.

At no time during the experiment or the training period was the purpose of the experiment revealed to participants. The purpose of this was to ensure that participants would not explicitly use a strategy that could lead to an alteration in their neural activation patterns in the different task conditions. Three of the six participants were members of the research group conducting the experiments, yet, our effects were reliable on a single-participant basis across all participants.

## Task: Spatial working memory mapping

Participants also performed an additional WM task while in the scanner, which served as training data for our classification analyses (see *Analysis: Spatial position decoding*). Identical to the main WM task, each trial began with the fixation point briefly turning green (750 ms), followed by a spatial memory target item (500 ms) at a random position (at 7° from fixation). The disappearance of the target was followed by a 12-s delay period, after which a white probe dot (radius = 0.15°) appeared at a random position (independent from the target position, also at 7° from fixation). Participants moved this probe dot around an invisible circle to match the position at which the memory target had been presented. Participants used the four fingers of their right hand to press different buttons that corresponded to fast (120°/s, outer two fingers) or slow (40°/s, inner two fingers) movement of the probe dot in

a counterclockwise (left two fingers) or clockwise (right two fingers) direction. This response period lasted 3 s, during which participants were able to move the dot back and forth to adjust its position as much as they wished. Once the 3-s response period was over, the probe dot disappeared and participants had no further ability to change their response. The final position of the probe dot was taken as the participant's response, and no further (visual) feedback of the response was provided.

Each run of the spatial WM mapping task consisted of 20 trials, with trials separated by an inter-trial interval jittered randomly within a range of 1–5 s. The total run length was 406 s. Participants performed 10 runs of this task in total, collected during a single scanning session. Across all 200 trials of the task, the spatial position of memory targets took on 200 distinct values uniformly spaced within a 360° space (i.e. 1.8° apart). To ensure a fairly even sampling of this space within individual runs, we binned these 200 possible positions into 20 bins of 10 positions each and generated a sequence where each run sampled from each bin once. The random starting position of the probe dot on each trial was generated using an identical procedure, but independently of the memory targets, so that there was no association between the position of the spatial memory target and the probe start position. The absolute average angular error across six participants on this task was 7.0 ± 0.9°.

## Task: Spatial localizer

We ran a spatial localizer task for two purposes, namely, to identify voxels having spatial selectivity within the region of space spanned by the memory positions (see *Identifying ROIs*), and to serve as training data for our classification analyses (see *Analysis: Spatial position decoding*). In this task, participants viewed black and white checkerboard wedges flickering at a rate of 4 Hz. Wedges had a width of 15° (polar angle), spanned an eccentricity range of 4.4–9.3° (visual angle), and were positioned at 24 different positions around an imaginary circle. Possible wedge center positions were offset from the cardinal axes by 7.5° (i.e. a wedge was never centered on the horizontal or vertical meridian). Each run included four wedge presentations at each position, totaling 96 trials. The sequence of positions was random with the constraint that consecutively presented wedges never appeared in the same quadrant. Trials were 3 s each and were not separated by an inter-trial interval. The total run length was 313 s. During each run, participants performed a change-detection task at fixation, where they responded with a button press any time the fixation point increased or decreased in brightness. A total of 20 brightness changes occurred in each run, at times that were random with respect to trial onsets. The magnitude of brightness changes was adjusted manually at the start of each run to control the difficulty. Average detection performance (hit rate) was 76.7 ± 4.2%. Participants performed between 8 and 16 total runs of this task. For some participants, some of these runs were collected as part of a separate experiment.

## Task: Sensorimotor cortex localizer

Participants also performed a sensorimotor cortex localizer task in the scanner. Analogous to our use of the spatial localizer task, this data served a dual purpose: it was used to identify ROIs in motor and somatosensory cortex that were selective for contralateral index finger button presses (see *Identifying ROIs*), and as a training set for one of our classification analyses (see *Analysis: Action decoding*). Participants attended a black fixation point (0.2°), and responded to brief (1000 ms) color changes of the fixation point by pressing a button with their left or right index finger. The fixation dot changed to either magenta or cyan to indicate which finger should be used, and each color change was separated by an inter-trial interval randomly jittered in the range of 2–6 s. Each run was 319 s long, and included 60 total trials (i.e. 60 button presses), with 30 trials for each finger randomly interleaved. The color/ finger mapping was switched on alternating runs. Participants were instructed to respond as quickly as possible to each color change. Average performance on this task was 92.8 ± 3.0% correct, and average behavioral response time was 530 ± 23 ms. Each participant performed six runs of this task.

## Analysis: Univariate

In order to estimate an HRF for each voxel during each condition (*Figure 1D*, *Figure 1—figure supplement 1*) we used linear deconvolution. We constructed a finite impulse response model (*Dale, 1999*) that included a series of regressors for trials in each task condition: one regressor marking the onset of a spatial memory target item, followed by a series of temporally shifted versions of that regressor (to model the BOLD response at each subsequent time point in the trial). The model also included a

constant regressor for each of the 20 total runs. The data used as input to this GLM was z-scored on a voxel-by-voxel basis within runs (see *Pre-processing of MRI data*). Estimated HRFs for the two conditions were averaged across all voxels within each ROI. To evaluate whether the mean BOLD signal in each ROI differed significantly between conditions, we used a permutation test. First, for each ROI and timepoint, we computed a Wilcoxon signed rank statistic comparing the activation values for each participant from condition 1 to the activation values for each participant from condition 2. Then, we performed 1000 iterations of shuffling the condition labels within each participant (swapping the condition labels for each participant with 50% probability). We then computed a signed rank statistic from the shuffled values on each iteration. Finally, we computed a two-tailed p-value for each ROI and timepoint by computing the number of iterations on which the shuffled signed rank statistic was ≥ the real statistic, and the number of iterations on which the shuffled statistic was ≤ the real statistic, and taking the smaller of these two values. We obtained the final p-value by dividing this value by the number of iterations and multiplying by 2.

For the analysis in which we separated contralateral and ipsilateral response trials (*Figure 1—figure supplement 2*), we performed the same procedure described above, except that for each voxel, we separated its responses into trials where the correct behavioral response corresponded to the index finger contralateral or ipsilateral to the voxel's brain hemisphere. This resulted in a separate estimate of each voxel's HRF for contralateral and ipsilateral trials in each condition. We then averaged the HRFs from each task condition and response type (contralateral or ipsilateral) across all voxels in both hemispheres of each ROI. Finally, within each condition, we tested the difference between contralateral and ipsilateral trials with a signed rank test with permutation as described above.

## Analysis: Spatial position decoding

We used linear classification to measure representations of remembered spatial position information in each visual and sensorimotor ROI during the main WM task. Since we were interested in assessing the coding format of memory representations, we separately performed decoding using three different approaches. In the first approach (*Figure 2*, *Figure 2—figure supplement 1*), we trained the decoder on independent data measured while participants were *remembering* a given spatial position (see *Task: Spatial working memory mapping*). In the second approach (*Figure 4A*), we trained the decoder on independent data collected when participants were *perceiving* a physical stimulus at a given spatial position (see *Task: Spatial localizer*). In the final method (*Figure 2—figure supplement 2*), we used cross-validation to train and test our decoder using data from within each condition of the main WM task.

Before performing each of these classification methods, we first mean-centered the voxel activation pattern from each trial in each task by subtracting the mean across all voxels from each trial. Next, we binned all trials of the main WM task (see *Task: Main working memory*) and the spatial WM mapping task (see *Task: Spatial working memory mapping*) into eight angular position bins that each spanned 45°, with the first bin centered at 0°. Trials from the spatial localizer task (see *Task: Spatial localizer*) were binned into eight angular position bins that each spanned 60° (i.e. the bins were slightly overlapping and some wedge positions contributed to multiple bins; similar results were obtained using bins that were entirely non-overlapping). We then performed binary classification between pairs of bins that were 180° apart (see *Figure 2A*), using a linear classifier based on the normalized Euclidean distance (for more details see *Henderson and Serences, 2019*). This meant that we constructed four separate binary classifiers, each operating on approximately one-fourth of the data, with a chance decoding value of 50%. We then averaged across the four binary classifiers to get a single index of classification accuracy for each ROI and task condition.

For the results shown in *Figure 2* and *Figure 2—figure supplement 1*, the training set for these classifiers consisted of data from the spatial WM mapping task (averaged within a fixed time window during the delay period 4.8–9.6 s after trial onset), and the test set consisted of data from the main WM task (either averaged over a window of 8–12.8 s after trial onset, or at each individual TR following trial onset). For the results based on sensory-driven responses (*Figure 4A*), the training set consisted of data from the spatial localizer task (averaged over a window 3.2–5.6 s after stimulus onset), and the test set consisted of data from the main WM task (same time window defined above). For the within-condition analyses (*Figure 2—figure supplement 2*), the training and testing sets both consisted of data from a single condition in the main WM task (either informative or uninformative; same time

window defined above). Each binary classifier was cross-validated by leaving out two trials at a time for testing (leaving out one trial per class ensures the training set was always balanced), and looping over cross-validation folds so that every trial served as a test trial once.

To test whether decoding performance was significantly above chance in each ROI and condition of the main WM task, we used a permutation test. On each of 1000 iterations, we shuffled the binary labels for the training set, trained a classifier on this shuffled data, and then computed how well this decoder predicted the binary labels in the test set. For each iteration, we then computed a Wilcoxon signed rank statistic comparing the N participants' real decoding values to the N participants' shuffled decoding values. A signed rank statistic greater than 0 indicated the median of the real decoding values was greater than the median of the shuffled decoding values, and a statistic less than zero indicated the median of the null decoding values was greater than the median of the real values. We obtained a one-tailed p-value for each ROI and task condition across all participants by counting the number of iterations on which the signed rank statistic was less than or equal to zero, and dividing by the total number of iterations.

To test whether decoding performance differed significantly between the two task conditions within each ROI, we used a permutation test. First, for each ROI, we computed a Wilcoxon signed rank statistic comparing the N participants' decoding values from condition 1 to the N participants' decoding values from condition 2. Then, we performed 1000 iterations of shuffling the condition labels within each participant (swapping the condition labels for each participant with 50% probability). We then computed a signed rank statistic from the shuffled values. Finally, we computed a two-tailed p-value for each ROI by computing the number of iterations on which the shuffled signed rank statistic was ≥ the real statistic, and the number of iterations on which the shuffled statistic was ≤ the real statistic, and taking the smaller of these two values. We obtained the final p-value by dividing this value by the total number of iterations and multiplying by 2.

The above procedures were used for all time-averaged and time-resolved decoding analyses (i.e. for time-resolved analyses we repeated the same statistical procedures at each timepoint separately). For the time-averaged decoding accuracies, we also performed a two-way repeated measures ANOVA with factors of ROI, condition, and a ROI × condition interaction (implemented using ranova.m). We performed a permutation test where we shuffled the decoding scores within each participant 1000 times, and computed an F-statistic for each effect on the shuffled data. Across all permutations, we obtained a null-distribution of F-values for effects of ROI, condition, and the ROI × condition interaction. The final p-values for each effect were based on the number of times the shuffled F-statistic for that effect was greater than or equal to the real F-statistic, divided by the total number of iterations (similar to method used in *Rademaker et al., 2019*). F-statistics reported in the text reflect the F-statistic obtained using the real (unshuffled) data.

## Analysis: Preview disk orientation decoding

To evaluate whether the difference between our task conditions was specific to the memory representations or a more global difference in pattern signal-to-noise ratio, we performed linear decoding of the orientation of the preview disk stimulus (*Figure 2—figure supplement 3*). The disk stimulus (*Figure 1A*) consisted of a light and a dark half, separated by a linear boundary that could range in orientation from 0 to 180° (for this analysis, we ignored the coloration of the disk stimulus, meaning which side was light gray and which was dark gray, and focused only on the orientation of the boundary). To run the decoding analysis, we first binned all trials according to the boundary orientation, using four bins that were centered at 0, 45, 90, and 135°, each bin 45° in size. We then trained and tested two separate linear decoders: one that classified the difference between the 0 and 90° bins, and one that classified the difference between the 45 and 135° bins. The final decoding accuracy value was the average of the accuracy from the two individual classifiers. This decoding approach is conceptually similar to that used for the spatial decoding analysis (*Figure 2A* and *Analysis: Spatial position decoding*) and also similar to that used by *Rademaker et al., 2019* for orientation decoding. The classifiers were always trained on data from one task condition at a time, and cross-validated by leaving one session of data out at a time (train on session 1, test on session 2, or vice versa). To capture the peak response to the preview disk presentation, for this analysis we used data averaged over a time window from 4.8 to 9.6 s after the trial start (note the preview disk appeared 3.5 s into the trial). As in the spatial decoding analyses, we mean-centered the voxel activation pattern for each

trial by subtracting the mean activation across voxels before performing the classification analysis. All statistical testing for the results of disk orientation decoding was done in an identical manner to the method described for spatial decoding accuracy (see *Analysis: Spatial position decoding*).

### Analysis: Action decoding

We performed linear classification (as above) to measure the representation of information related to left or right index finger button presses in each ROI. To assess the coding format of action representations, we separately performed decoding using two different approaches. In the first approach, we used data from the main WM task to train and test the classifier (*Figure 3*, *Figure 3—figure supplement 1*). Here, action classification was always done using data from one task condition at a time (i.e. informative or uninformative trials). The decoder was always trained on data from one session and tested on the other session. Because the mapping of disk side color (light or dark gray) to finger was always switched between the two sessions, this ensured that the information detected by the classifier was not related to the luminance of the half of the disk corresponding to the response finger. In the second approach, we trained the classifier using data from a separate task during which participants were *physically pressing a button* (see *Task: Sensorimotor cortex localizer*), and we tested using data from the main WM task (*Figure 4B*). Irrespective of the training-testing approach used, classification was based on trials labeled according to the finger (left or right index) that corresponded to the correct response. This means that all trials were included (also those where the incorrect button, or no button, was pressed), which ensured that the training set for the classifier was balanced. Note that qualitatively similar results were obtained when using correct trials only, or when using the participant's actual response as the label for the decoder.

The above procedure was used for both time-averaged (*Figure 3A* and *Figure 4B*) and time-resolved action decoding (*Figure 3B*, *Figure 3—figure supplement 1*). For time-averaged decoding, single trial responses were obtained by averaging over specified time windows after trial onset (8–12.8 s for the main WM task; 3.2–5.6 s for the sensorimotor cortex localizer). For time-resolved decoding on the main WM task data, the training and testing set each consisted of data from the TR of interest (but from different sessions, as described above). All statistical tests on the results of action decoding were performed in an identical manner to the statistics on the results of spatial decoding (see *Analysis: Spatial position decoding*).

## Acknowledgements

Funded by NEI R01-EY025872 to JS, NIMH Training Grant in Cognitive Neuroscience (T32-MH020002) to MH, and a European Union's Horizon 2020 research and innovation program under the Marie Sklodowska-Curie Grant Agreement No 743,941 to RR and NIH-MH087214. We thank Kirsten Adam, Timothy Sheehan, and Sunyoung Park for helpful discussions and data collection.

## Additional information

#### Competing interests

John T Serences: Reviewing editor, eLife. The other authors declare that no competing interests exist.

#### Funding

| Funder | Grant reference number | Author |
| --- | --- | --- |
| National Eye Institute | R01-EY025872 | John T Serences |
| National Institutes of Health | T32-MH020002 | Margaret M Henderson |
| European Union Horizon 2020 Research and Innovation Program | Marie Sklodowska-Curie Grant Agreement No 743941 | Rosanne L Rademaker |
| National Institutes of Health | MH087214 | John T Serences |

| Funder | Grant reference number | Author |
|--------|------------------------|--------|

The funders had no role in study design, data collection and interpretation, or the decision to submit the work for publication.

## Author contributions

Margaret M Henderson, Conceptualization, Data curation, Formal analysis, Investigation, Methodology, Software, Visualization, Writing – original draft, Writing – review and editing; Rosanne L Rademaker, Conceptualization, Investigation, Methodology, Software, Writing – review and editing; John T Serences, Conceptualization, Funding acquisition, Methodology, Project administration, Resources, Software, Supervision, Writing – review and editing

## Author ORCIDs
Margaret M Henderson (ID) http://orcid.org/0000-0001-9375-6680

## Ethics

Human subjects: The study protocol was approved by the Institutional Review Board at UCSD (approval code 180067), and all participants provided written informed consent and consent to publish.

## Decision letter and Author response
Decision letter https://doi.org/10.7554/eLife.75688.sa1
Author response https://doi.org/10.7554/eLife.75688.sa2

# Additional files

## Supplementary files
• Supplementary file 1. Table listing the number of voxels in each ROI for each participant and hemisphere. Sizes of retinotopic visual ROIs (V1-IPS3) are after thresholding with a spatial localizer (see *Methods, Task: Spatial Localizer*). S1, M1, and PMc were defined using a button-pressing task (see *Methods, Task: Sensorimotor Cortex Localizer*). All analyses in this paper were done using bilateral ROIs (i.e. concatenating the left and right hemispheres of each ROI).

• Transparent reporting form

## Data availability
The dataset from this manuscript is publicly available on Open Science Framework at https://osf.io/te5g2/. All code associated with the manuscript is publicly available on GitHub at https://github.com/mmhenderson/wm_flex, (copy archived at swh:1:rev:a25485d9983bd13b99f99f98d24b0faeaff50005).

The following dataset was generated:

| Author(s) | Year | Dataset title | Dataset URL | Database and Identifier |
|-----------|------|---------------|-------------|-------------------------|
| Henderson MM, Rademaker RL, Serences JT | 2022 | Open Data 2022: Flexible utilization of spatial- and motor-based codes for the storage of visuo-spatial information | https://osf.io/te5g2/ | Open Science Framework, 10.17605/OSF.IO/TE5G2 |

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
