## [Editor Report]

This rigorous, carefully designed and executed functional magnetic-resonance imaging study provides compelling evidence against a rigid, fixed model for how working-memory representations are maintained in the human brain. By analyzing patterns and strength of brain activity, the authors show that networks for maintaining contents in mind vary depending on the task demands and foreknowledge of anticipated responses. This manuscript will be of interest to scientists studying working memory, both in humans and in non-human primates.

---

## [Decision Letter]

**Decision letter after peer review:**

Thank you for submitting your article "Flexible utilization of spatial-and motor-based codes for the storage of visuo-spatial information" for consideration by *eLife*. Your article has been reviewed by 2 peer reviewers, and the evaluation has been overseen by Chris Baker as the Senior and Reviewing Editor. The following individuals involved in the review of your submission have agreed to reveal their identity: Elizabeth Lorenc (Reviewer #1); Anna Christina Nobre (Reviewer #2).

Essential revisions:

The reviewers think the manuscript addresses an important question and are enthusiastic about many aspects of the study. However, both raise questions that have a potential impact on the interpretation. These concerns should be directly addressed in a revision (see below for reviewers' full comments).

In particular, please discuss/elaborate:

1) Why representations do not completely disappear from visual cortex when a button press can be planned in advance (R1).

2) Whether there may be a contribution of an actual finger movement (R1).

3) How the multiple differences between the informative x non-informative conditions affect the interpretation of the results (R2).

4) The potential contribution of anticipated difficulty, arousal, or other nuisance variables (R2).

5) The distinction between retrospective and prospective codes (R2).

*Reviewer #1 (Recommendations for the authors):*

This is a clear manuscript that was enjoyable to read!

*Reviewer #2 (Recommendations for the authors):*

1. One study that specifically speaks to motor readiness when responses are linked to items maintained in visual working memory is by van Ede, Chekroud, Stokes and Nobre 2019 Nature Neuroscience.

2. Reading between the lines, some of the few participants were experienced and perhaps non-naïve to the experimental questions? I think this may be worth stating.

3. When describing the task (in methods), it would be good to have a summary statement of what the participant's task actually was before stepping through the various events in the trial.

4. In figure 1, it seems counter-intuitive to place the spatial WM task (b) in between the main task (a) and its related behavioural results. Up to you, but maybe consider swapping order of (b) and (c) to have task and performance together.

5. A bit more detail on how participants reported the spatial location on the spatial WM mapping would be welcome. Could they correct/tinker with responses? Was there a confirmation response?

6. Similarly, was instruction for responding in core task about whether it was in 'lighter x darker' half? The text (results) says "to indicate on which of the two halves of the disk the target dot had been presented…" but it falls short of stating how the 2 halves were categorised (i.e., light x dark, counter x clockwise). I may have missed this somewhere.

---

## [Author Response]

Essential revisions:The reviewers think the manuscript addresses an important question and are enthusiastic about many aspects of the study. However, both raise questions that have a potential impact on the interpretation. These concerns should be directly addressed in a revision (see below for reviewers' full comments).In particular, please discuss/elaborate:1) Why representations do not completely disappear from visual cortex when a button press can be planned in advance (R1).

We address this point in our reply to R1 Public Review (major comment 1).

2) Whether there may be a contribution of an actual finger movement (R1).

We address this point in our reply to R1 Public Review (major comment 2).

3) How the multiple differences between the informative x non-informative conditions affect the interpretation of the results (R2).

We address this point in our reply to R2 Public Review (major comment 1).

4) The potential contribution of anticipated difficulty, arousal, or other nuisance variables (R2).

We address this point in our reply to R2 Public Review (major comment 2).

5) The distinction between retrospective and prospective codes (R2).

We address this point in our reply to R2 Public Review (major comment 3).

Reviewer #2 (Recommendations for the authors):1. One study that specifically speaks to motor readiness when responses are linked to items maintained in visual working memory is by van Ede, Chekroud, Stokes and Nobre 2019 Nature Neuroscience.

Yes, we agree this study is very relevant to our work. We had cited it only briefly in the original version of our manuscript, and we have now added an additional discussion of the work towards the end of page 14.

2. Reading between the lines, some of the few participants were experienced and perhaps non-naïve to the experimental questions? I think this may be worth stating.

We have added a note about this to our methods section (page 22). All our participants were naïve to the experimental question. Three out of six were lab members who may have guessed at the purpose of the task manipulation, but the experimental question was not disclosed to them until after their participation. The other three participants were not lab members (though they were experienced MRI participants), and unlikely to have given much thought to the experimental question. Our effects were reliable on a single-participant basis across all participants (i.e., see single gray lines in Figure 2B).

3. When describing the task (in methods), it would be good to have a summary statement of what the participant's task actually was before stepping through the various events in the trial.

We have added this to our methods section (page 20).

4. In figure 1, it seems counter-intuitive to place the spatial WM task (b) in between the main task (a) and its related behavioural results. Up to you, but maybe consider swapping order of (b) and (c) to have task and performance together.

Good suggestion. We have switched the order of panels (b) and (c) in Figure 1.

5. A bit more detail on how participants reported the spatial location on the spatial WM mapping would be welcome. Could they correct/tinker with responses? Was there a confirmation response?

We have added some additional details on this task report method to the methods section (page 22). It now states that: “This response period lasted 3 seconds, during which participants were able to move the dot back-and-forth to adjust its position as much as they wished. Once the 3 second response period was over, the probe dot disappeared and participants had no further ability to change their response. The final position of the probe dot was taken as the participant’s response, and no further (visual) feedback of the response was provided. “

6. Similarly, was instruction for responding in core task about whether it was in 'lighter x darker' half? The text (results) says "to indicate on which of the two halves of the disk the target dot had been presented…" but it falls short of stating how the 2 halves were categorised (i.e., light x dark, counter x clockwise). I may have missed this somewhere.

We have added some additional details on this aspect of the task instructions to the methods section (page 21). This now states that: “Following the delay period, a second disk stimulus appeared for 2 seconds, serving as the response probe (i.e., the “response disk”). At this point, participants responded with a button press to indicate which side of the response disk the memory target had been presented on. Specifically, they used either their left or right index finger to indicate whether the target dot fell within the light gray or dark gray side of the response disk. On each scan run, the light gray shade and dark gray shade were each associated with one response finger, and the mapping between shade of gray (light or dark) and finger (left or right index finger) was counter-balanced across sessions within each participant. Participants were reminded of the mapping between shades of gray and response fingers at the start of each scan run.”